# Phenotypes of non-alcoholic fatty liver disease (NAFLD) and all-cause mortality: unsupervised machine learning analysis of NHANES III

Rodrigo M Carrillo-Larco [1,2,3,4] Wilmer Cristobal Guzman-Vilca [3,5,6]
Manuel Castillo-Cara [7] Claudia Alvizuri-Gómez,[8] Saleh Alqahtani,[9,10]
Vanessa Garcia-Larsen [11]

**Correspondence to**
Dr Vanessa Garcia-Larsen;
vgla@jhu.edu

## ABSTRACT

**Objectives** Non-alcoholic fatty liver disease (NAFLD) is a non-communicable disease with a rising prevalence worldwide and with large burden for patients and health systems. To date, the presence of unique phenotypes in patients with NAFLD has not been studied, and their identification could inform precision medicine and public health with pragmatic implications in personalised management and care for patients with NAFLD.

**Design** Cross-sectional and prospective (up to 31 December 2019) analysis of National Health and Nutrition Examination Survey III (1988–1994).

**Primary and secondary outcomes measures** NAFLD diagnosis was based on liver ultrasound. The following predictors informed an unsupervised machine learning algorithm (k-means): body mass index, waist circumference, systolic blood pressure (SBP), plasma glucose, total cholesterol, triglycerides, liver enzymes alanine aminotransferase, aspartate aminotransferase and gamma glutamyl transferase. We summarised (means) and compared the predictors across clusters. We used Cox proportional hazard models to quantify the all-cause mortality risk associated with each cluster.

**Results** 1652 patients with NAFLD (mean age 47.2 years and 51.5% women) were grouped into 3 clusters: anthro-SBP-glucose (6.36%; highest levels of anthropometrics, SBP and glucose), lipid-liver (10.35%; highest levels of lipid and liver enzymes) and average (83.29%; predictors at average levels). Compared with the average phenotype, the anthro-SBP-glucose phenotype had higher all-cause mortality risk (aHR=2.88; 95% CI: 2.26 to 3.67); the lipid-liver phenotype was not associated with higher all-cause mortality risk (aHR=1.11; 95% CI: 0.86 to 1.42).

**Conclusions** There is heterogeneity in patients with NAFLD, whom can be divided into three phenotypes with different mortality risk. These phenotypes could guide specific interventions and management plans, thus advancing precision medicine and public health for patients with NAFLD.

## INTRODUCTION

The epidemiology of non-communicable diseases (NCDs) is largely driven by cardiometabolic risk factors and diseases,

## STRENGTHS AND LIMITATIONS OF THIS STUDY

⇒ We leveraged on population-based data with image-based non-alcoholic fatty liver disease (NAFLD) diagnosis and long-term follow-up.

⇒ We applied machine learning techniques to identify phenotypes among people with NAFLD, consistent with precision medicine and precision public health.

⇒ Arguably, the gold standard for NAFLD diagnosis is liver biopsy, though given the source of the dataset (population-based nationally representative study), taking biopsies was not possible.

⇒ We did not use diet-related variables because measuring these may vary between places and observers unlike blood biomarkers.

namely dyslipidaemias, type 2 diabetes mellitus (T2DM), hypertension and cardiovascular diseases. Nonetheless, there are other NCDs rapidly growing along with, and as a consequence of,[1 2] the afore-mentioned cardiometabolic conditions. Non-alcoholic fatty liver disease (NAFLD) is an outstanding example with a soaring prevalence worldwide,[3–6] high economic costs for patients and health systems,[7 8] and poor patient-oriented outcomes.[9] Despite this high disease burden, there are no specific treatments for NAFLD other than managing the underlying conditions such as obesity or T2DM together with recommendations for healthy lifestyles.[10–14] Consistent with the global call for precision medicine and public health, a way to maximise the benefits of available and forthcoming[15–17] treatments for NAFLD, could be to identify phenotypes in patients with NAFLD. Therefore, specific management plans can be proposed according to the underlying profile of each phenotype. A data-driven machine learning (ML) approach has been proven to be a reliable method to identify phenotypes among patients, and there are

several examples for T2DM.[18] However, NAFLD research has not yet used ML to identify phenotypes or subpopulations, even though it could signal groups of patients for which different management plans can be provided and different prognosis can be expected. We aimed to identify phenotypes in people with NAFLD in the general population and to quantify the all-cause mortality risk associated with each phenotype.

## METHODS

### Study design and data sources

This is a data-driven analysis following an unsupervised ML approach. We conducted a cross-sectional analysis whereby we identified clusters of patients with NAFLD and described the underling phenotype of each cluster. Analysing the same study population, we also conducted a prospective analysis whereby we investigated whether the phenotypes were associated with higher all-cause mortality risk. Overall, for both the cross-sectional and prospective analyses, we used the same study population from the National Health and Nutrition Examination Survey (NHANES) III.

We used individual-level data from the NHANES III conducted between 1988 and 1994. The NHANES III included a nationally representative sample of non-institutionalised individuals in the USA.[19] We used NHANESS III, and no more recent iterations of this survey, because NHANESS III is the only with liver ultrasound data, which provide high-quality information for NAFLD research.

### Study population

The study population included people aged between 20 and 74 years who had hepatic imaging data. Hepatic imaging refers to ultrasound examinations conducted following standard procedures to secure consistent and reliable results for all participants.[20] We included people whose imaging results were deemed 'confident' or 'absolute' to secure high-quality data for the definition of NAFLD.[20] For the analyses, we only included people with NAFLD defined as a 'moderate–severe' hepatic steatosis.[20] In so doing, we excluded missing observations in liver ultrasound data; missing patterns in this variable were not explored.

People with a positive test for hepatitis B and hepatitis C were excluded and so were people with high alcohol consumption (online supplemental materials p. 02). People whose risk factors levels were below or above these plausibility thresholds were excluded too: body mass index (BMI) between 10 and 80 kg/m$^2$, waist circumference between 30 and 200 cm, systolic blood pressure (SBP) between 70 and 270 mm Hg, plasma glucose between 45 and 540 mg/dL, total cholesterol between 20 and 773 mg/dL and triglycerides between 17 and 1771 mg/dL. Details about data preparation and selection of the study population are available in online supplemental materials p. 02.

### Variables

From the NHANES dataset, we selected 10 predictors, which will inform the composition of the clusters and characterise the phenotype of each cluster. These predictors were selected because they are established cardiometabolic risk factors, and cardiovascular diseases are a leading cause of mortality in people with NAFLD.[21] We also included liver biomarkers as predictors because NAFLD is a chronic liver condition. The predictors included chronological age (years), BMI (kg/m$^2$) based on measured weight and height, waist circumference (cm), SBP (mm Hg), plasma glucose (mg/dL), which was used regardless of fasting duration (41.53% or 686/1652 had a fasting duration <8 hours) to maximise sample size for the study population comprising only people with NAFLD, total cholesterol (mg/dL), triglycerides (mg/dL), serum alanine aminotransferase (U/L), serum aspartate aminotransferase (U/L) and serum gamma glutamyl transferase (GGT; U/L). All blood biomarkers were collected and analysed the following standard procedures.[22] We conducted a completed-case analysis for the 10 predictors of interest.

We did not include more predictors, such as coagulation or anaemia biomarkers, to deliver a model that can be replicated in many places and by researchers who may not have access to more sophisticated biomarkers. Including more or more sophisticated predictors would limit the use of our model to places where these predictors are available. This would, most likely, exclude several primary healthcare facilities and those in rural or resource-limited settings.

Mortality data are also available for NHANES III.[23] The National Centre for Health Statistics delivers a mortality dataset linked to NHANES participants (adults only). Follow-up duration, in person months, is computed from the examination date to the date of death or censorship (31 December 2019). Because a priori we did not know the number of clusters or their underlying profiles, we focused on all-cause mortality rather than on a specific cause of death. For example, had the unsupervised analysis suggested many clusters, it would have compromised the CIs from the regressions if the exposure variable was divided in several clusters or groups.

### Analysis

#### Open-access resources

The analyses were conducted in Python V.3.10 using PyCharm V.2021.3.3 and Jupyter Notebooks; we also used R V.4.1.2 for the statistical analyses. The Jupyter Notebooks and R scripts are available as online supplemental material. All analytical methods are detailed in online supplemental materials pp. 02–05.

#### Unsupervised ML analysis

We applied principal components analysis to the 10 predictors. We chose four principal components because these explained 95.19% of the variance. In other words, we retained 95.19% of the information in the original

**Table 1** General characteristics of participants by phenotype cluster

| Clusters | Age | BMI (kg/m²) | Waist (cm) | SBP (mm Hg) | Glucose (mg/dL) | TC (mg/dL) | Triglycerides (mg/dL) | Aspartate (U/L) | Alanine (ALT) | GGT (U/L) |
|---|---|---|---|---|---|---|---|---|---|---|
| Average | 46.6 (15.9) | 29.9 (6.4) | 100.0 (15.7) | 127.9 (19.6) | 99.3 (18.3) | 205.9 (42.6) | 149.7 (72.7) | 23.6 (14.1) | 21.9 (17.1) | 33.7 (28.0) |
| Lipid-liver | 47.1 (13.9) | 30.2 (4.7) | 103.3 (11.5) | 132.5 (18.3) | 114.9 (43.1) | 241.0 (55.8) | 495.7 (221.1) | 34.6 (40.5) | 36.5 (44.2) | 93.0 (133.2) |
| Anthro-SBP-glucose | 55.3 (12.1) | 31.7 (6.7) | 106.1 (13.8) | 135.9 (19.5) | 288.4 (72.2) | 232.9 (64.0) | 248.0 (147.7) | 20.7 (14.8) | 22.4 (17.9) | 48.0 (51.5) |

All variables are expressed as means (SD). Comparisons across clusters are shown in figure 1 and online supplemental table 1.
Glucose refers to plasma glucose, aspartate refers to aspartate aminotransferase, alanine refers to alanine aminotransferase.
BMI, body mass index; GGT, gamma-glutamyl transferase; SBP, systolic blood pressure; TC, total cholesterol.

dataset with these four components. On these four principal components, we applied the k-means algorithm, an unsupervised ML method, to identify clusters of patients with NAFLD with unique phenotypes. To define the number of clusters, which is unknown at the beginning of any unsupervised ML analysis, we followed a data-driven approach together with expert knowledge about NAFLD. First, we plotted a dendrogram with Euclidean distances, which suggested to use five clusters. Second, the Elbow method suggested to use five or six clusters. Third, the Silhouette Score was highest with two clusters, followed by three clusters. Fourth, the Jaccard Score was highest for three clusters. Therefore, based on the above metrics and expert knowledge, we decided on three clusters. In exploratory analyses, we also tried selecting the number of clusters based on other algorithms and the results were consistent with those herein summarised; the other algorithms we tried were: balanced iterative reducing and clustering using hierarchies, spectral and agglomerative.

## Statistical analysis

First, we conducted a descriptive analysis to characterise the 3 clusters in terms of the 10 predictors. That is, we summarised the mean of the 10 predictors in each cluster and used Boxplots. We used t-tests for pair-wise comparisons between all clusters; the p value is reported accounting for the Bonferroni correction for multiple comparisons.

Second, to study the all-cause mortality associated with each cluster, we ran Cox proportional hazard models reporting HRs along with the 95% CI. We used time in the study (difference between when the one-off baseline evaluation occurred and the endpoint) as the time scale in the Cox models. We fitted three models: (1) crude model only including the outcome (all-cause mortality) and the exposure (clusters, categorical variable with three levels); (2) model 1, including sex; and (3) model 2, including sex, smoking status (current smoker of cigarettes, no vs yes), years of education and household income (<US$20 000 vs ≥US$20 000 in the last year). The proportional hazard assumption was verified in the crude model, and we did not observe evidence that this assumption was violated. We did not adjust for age or other cardiometabolic risk factors because these were included in the clusters, which are the main predictors (ie, independent variable) in the regression models. Together with the Cox regression model, we also report the absolute number of deaths per group, the total follow-up duration per group, as well as the incidence rate per 1000 person-years, which was computed with the epi.conf function from the epiR package in R.

The statistical analysis did not account for the complex survey design of the NHANES survey (eg, sampling weights). We aimed to characterise the clusters and study if they were associated with higher all-cause mortality risk, rather than to provide prevalence estimates or means that are representative at the national level.

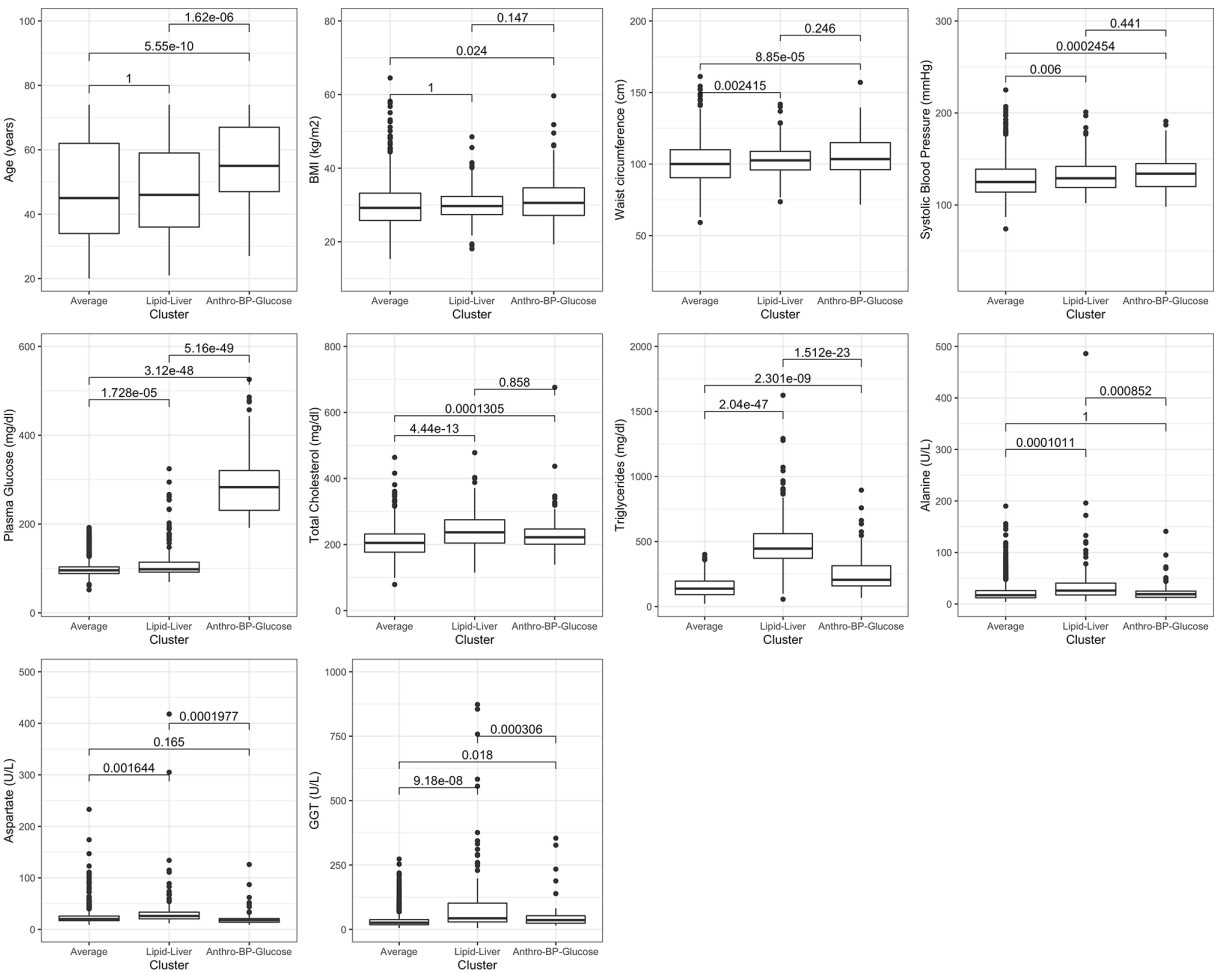

**Figure 1** Distribution of the predictors by phenotype cluster. P values are for pair-wise comparisons and Bonferroni adjusted for multiple comparisons (online supplemental table 1). BMI, body mass index (kg/m$^2$); GGT, gamma-glutamyl transferase (U/L).

## Sensitivity analysis

We carried several sensitivity analyses to elucidate if any observed differences were driven by the fact that we did not only include people with more than 8 hours of fasting. First, we compared the mean plasma glucose levels across phenotypes restricting the sample to those with eight or more hours of fasting duration. Second, we ran the fully adjusted Cox model (model 2) restricting the sample to those with eight or more hours of fasting. Third, we ran the fully adjusted Cox model (model 2) including fasting duration (in hours) as an additional confounder rather than restricting the sample to those with eight or more hours of fasting duration. To explore whether the phenotypes were associated with all-cause mortality above and beyond the individual risk factors, we also ran model 2 including the 10 predictors as confounders.

## Patient and public involvement

It was not appropriate or possible to involve patients or the public in the design, or conduct, or reporting, or dissemination plans of our research.

## RESULTS
### Study population

There were 1652 people with NAFLD, the mean age was 47.2 years and 51.5% were women. Across the study population, the mean BMI was 30.0 kg/m$^2$ and the mean waist circumference was 100.7 cm; the mean SBP was 128.9 mm Hg, the mean plasma glucose was 112.9 mg/dL, the mean total cholesterol was 211.3 mg/dL and the mean triglycerides was 191.8 mg/dL; the mean aspartate was 24.5 U/L, the mean alanine was 23.5 U/L and the mean GGT was 40.7 U/L.

### Cluster profiles

Given the underlying profiles of each phenotype (table 1 and figure 1), these were labelled as it follows: lipid-liver (ie, highest levels of cholesterol and liver enzymes), anthro-SBP-glucose (ie, highest levels of anthropometrics, SBP and glucose) and average (ie, mean levels of the predictors were always in between the former two phenotypes). The average phenotype grouped 1376 people (83.29%), followed by the lipid-liver (171 people, 10.35%) and the anthro-SBP-glucose phenotype with 105 people (6.36%).

**Table 2** Mortality risk in patients with non-alcoholic fatty liver disease by phenotype cluster

| | | | | Outcome: all-cause mortality (HR, 95% CI)* | | |
| | Absolute number of deaths | Total follow-up duration (years) | Incidence rate per 1000 person-years (95% CI) | Crude N=1651 Events=678 | Model 1 N=1651 Events=678 | Model 2 N=1622 Events=663 |
| Cluster | | | | | | |
|---|---|---|---|---|---|---|
| Average | 529 | 31 707.083 | 16.7 (15.3 to 18.2) | 1 | 1 | 1 |
| Lipid-liver | 71 | 3685.417 | 19.3 (15.1 to 24.3) | 1.17 (0.91 to 1.50) | 1.11 (0.86 to 1.42) | 1.11 (0.86 to 1.42) |
| Anthro-SBP-glucose | 78 | 1766.000 | 44.2 (34.9 to 55.1) | 2.94 (2.32 to 3.74) | 2.97 (2.34 to 3.78) | 2.88 (2.26 to 3.67) |

In the average group, there was one observation with missing data on follow-up duration.
*Cox proportional HRs; model 1 included sex. Model 2 included, in addition to sex, smoking status (current, no vs yes); years of education (numeric variable); and household income (<US$20 000 vs ≥US$20 000 in the last year). For the crude model, the global test for the proportional-hazard assumption was not significant (p=0.078), suggesting this assumption holds, which is also supported by the Schoenfeld residuals plot (online supplemental figure 2). The exposure variable for all models (crude, model 1 and model 2) was a three-level variable derived from the cluster analysis; in other words, the cluster analysis derived a new variable with three levels which we called phenotypes (average, lipid-liver and anthro-SBP-glucose) whose profiles were described in figure 1. Neither of the three models included the predictors used to derive the clusters or phenotypes. The predictors were used to derive and characterise the phenotypes only (not to adjust the models).

Differences between the average and the lipid-liver phenotypes were always statistically significant, except for age (p=1) and BMI (p=1) (figure 1 and online supplemental table 1). Differences between the average and anthro-SBP-glucose phenotypes were always statistically significant except for alanine (p=1) and aspartate (p=0.165). Differences between the lipid-liver phenotype and the anthro-SBP-glucose phenotype were significant for six predictors: age (older in the anthro-SBP-glucose), plasma glucose (higher in the anthro-SBP-glucose), triglycerides (higher in the lipid liver), alanine (higher in the lipid liver), aspartate (higher in the lipid liver) and GGT (higher in the lipid liver). That is, most of the unique features of the lipid-liver phenotype were different—higher—in comparison to the anthro-SBP-glucose phenotype; conversely, only glucose in the anthro-SBP-glucose phenotype was significantly different—higher— relative to the lipid-liver phenotype.

## All-cause mortality
Among those who died, 78.0% were in the average phenotype, 10.5% were in the anthro-SBP-glucose phenotype and 11.5% were in the lipid-liver phenotype. The mean follow-up duration was 270.1 months (~22.5 years), ranging from a month to 373 months. The survival probability decreased much faster for the anthro-SBP-glucose phenotype than for the lipid-liver phenotype (online supplemental figure 1).

The crude Cox proportional hazard model showed that, in comparison to the average phenotype, the lipid-liver was not associated with higher all-cause mortality (HR=1.17; p=0.217); conversely, the anthro-SBP-glucose phenotype was associated with higher all-cause mortality risk (HR=2.94; p<0.001; table 2). After adjusting for sex, the association for the lipid-liver phenotype was still not significant (HR=1.11; p=0.431), and the association for the anthro-SBP-glucose phenotype remained significant (HR=2.97; p<0.001). The same pattern was observed when we further adjusted for smoking, education and household income whereby the lipid-liver phenotype was not associated with higher all-cause mortality risk (HR=1.21;

p=0.427), yet the anthro-SBP-glucose phenotype was still strongly associated with higher all-cause mortality risk (HR=2.88; p<0.001).

## Sensitivity analyses
The anthro-SBP-glucose phenotype still had the highest mean plasma glucose (273.5 mg/dL), followed by the lipid-liver phenotype (115.9 mg/dL) and the average phenotype (101.2 mg/dL). The p value for the Bonferroni-adjusted t-test for all pair-wise comparisons was significant (p<0.003).

When model 2 for the Cox regression was restricted to people with eight or more hours of fasting, the results were consistent with the main analysis: the anthro-SBP-glucose phenotype was strongly associated with higher all-cause mortality risk (HR=2.65; 95% CI: 1.90 to 3.70; p<0.001), and the association for the lipid-liver phenotype was not significant (HR=1.12; 95% CI: 0.80 to 1.58; p=0.512). Virtually the same results were observed when model 2 included fasting duration as a covariate rather than restricting the sample to those with eight or more hours of fasting duration: HR=2.89 (95% CI: 2.27 to 3.69; p<0.001) for the anthro-SBP-glucose and HR=1.11 (95% CI: 0.86 to 1.43; p=0.414) for the lipid-liver phenotype.

When model 2 also included the 10 predictors used to define the phenotypes, neither the lipid-liver (HR=1.20; 95% CI: 0.82 to 1.74; p=0.342) phenotype nor the anthro-SBP-glucose (HR=1.46; 95% CI: 0.90 to 2.38; p=0.126) phenotype was associated with higher all-cause mortality risk.

## DISCUSSION
### Main results
In a sample of patients with NAFLD from the general population and using simple predictors, a ML analysis revealed three phenotypes with unique profiles. The anthro-SBP-glucose phenotype had the highest mean levels of BMI, waist circumference, SBP and plasma glucose. Patients in the lipid-liver phenotype had, on average, the highest levels of cholesterol and liver biomarkers. There was a

third phenotype whose members had average risk factor levels. Most patients with NAFLD belonged to the average phenotype (83%), followed by the lipid-liver (10%) and anthro-SBP-glucose (6%) phenotypes. In comparison to the average phenotype, the anthro-SBP-glucose phenotype had higher all-cause mortality risk; the all-cause mortality risk appeared not to be different between the average and lipid-liver phenotypes.

We believe that these findings have the potential to advance precision medicine and public health by identifying variations in the clinical and metabolic presentation of NAFLD. Such variations suggest three homogenous phenotypes. Patients with NAFLD with the anthro-SBP-glucose phenotype, which was associated with the highest all-cause mortality risk, should receive urgent counselling and medication to reduce weight, SBP and glucose levels. While maintaining a healthy lifestyle should always be a cornerstone in the management of patients with NAFLD, those with the lipid-liver phenotype could most benefit from novel therapies targeting cholesterol and liver physiology.[15–17] The average phenotype was the most frequent. This signals a group of patients with NAFLD who should be carefully monitored, though, perhaps, not with the same frequency or intensity as for the other phenotypes in which risk factors levels were higher and had higher all-cause mortality risk.

### Results in context

NAFLD, defined as >5% of hepatic steatosis in images or histology in the absence of secondary causes (eg, viral hepatitis),[11] is the most common cause of chronic liver disease in the last decade.[24] Although NAFLD is distributed worldwide, the prevalence of NAFLD is particularly high in the Middle East and South America.[24] Future research in these world regions could elaborate on our work to trial specific recommendations according to the patient's cluster.

To date, there are no specific treatments to prevent or cure NAFLD. For example, vitamin E has been proposed as a potential therapy for NAFLD because it may improve non-alcoholic steatohepatitis. However, systematic reviews and meta-analysis of randomised trials have shown inconsistent evidence to recommend vitamin E for NAFLD[25]; moreover, potential adverse effects of vitamin E should be taken into consideration.[14] Although other vitamins may play a role as well, to the best of our knowledge, their effect has not been summarised in meta-analysis of randomised control trials (high level of evidence) and is less elaborated in clinical guidelines. Current strategies focus on weight loss, exercise and diet. Mediterranean diet appears to reduce hepatic steatosis due to reduction in insulin resistance and total lipid concentrations[26]; likewise, weight loss of ≥7% correlates with improvement of histological features.[14] Overall, implementing these interventions to improve healthy lifestyles should be the standard of care for patients with NAFLD, and more so for those in the anthro-SBP-glucose phenotype herein found.

In 2020, an international consensus proposed the term metabolic-associated fatty liver disease (MAFLD) to substitute NAFLD due to the strong metabolic component of this chronic liver condition.[27] Our work may support the MAFLD nomenclature because the anthro-SBP-glucose phenotype, largely consisting of traits consequence of metabolic dysfunction, had the highest all-cause mortality risk. Nevertheless, our work also signals there is heterogeneity in patients with NAFLD, which may better guide treatment options above and beyond one specific nomenclature (NAFLD vs MAFLD).

Few studies have quantified the association between phenotypes and mortality in patients with NAFLD. Golabi *et al*[28] evaluated the effect of abdominal and overall adiposity on the mortality risk in patients with NAFLD; after 22.4 years of follow-up, they observed that patients with NAFLD and normal BMI but high waist circumference had higher risk of cardiovascular mortality.[28] This is consistent with the anthro-SBP-glucose phenotype having the highest all-cause mortality risk. It is worth noting that, even though on average the anthro-SBP-glucose phenotype had high levels of anthropometrics, SBP and glucose, it is not the same as a dichotomous variable whereby people with high anthropometrics, SBP and glucose were on one or the other category of the dichotomous variable (similar to the definition of metabolic syndrome); nevertheless, the additive effect of the anthropometrics, SBP and glucose should be noted and could potentially explain the higher mortality.

### Strengths and limitations

We leveraged on population-based data with image-based[20] NAFLD diagnosis and long-term follow-up.[23] We applied ML techniques to identify phenotypes among people with NAFLD, consistent with precision medicine and precision public health. Not only did we describe the underlying profile of each phenotype but we also quantified the all-cause mortality risk associated with each phenotype.

Notwithstanding, there are limitations we should acknowledge. First, the gold standard to diagnose NAFLD requires a liver biopsy. Collecting a liver biopsy from a large population-based sample as that of NHANES would be (almost) impossible. From a clinical perspective, the fact that NAFLD was diagnosed with liver ultrasound could be a limitation; however, from a global or population health perspective, it could be a step forward to characterise phenotypes in people with NAFLD. Second, aiming to deliver a friendly and easy to replicate model, we included ten predictors that may be accessible to many clinicians and researchers. Had we used more sophisticated predictors, it would limit the reproducibility of our model by a broad audience. However, this decision could have led to the omission of other potential phenotypes. For example, if we had included anaemia biomarkers (eg, haemoglobin or ferritin), there could have been an additional phenotype characterised by anaemia. Our work opens the door to use ML in NAFLD research aiming for

precision medicine and public health, and future work could expand our set of phenotypes for specific settings where access to these predictors is not a limitation. In this line, although diet is a key variable in NAFLD, we did not include diet-based variables (including alcoholic beverages) in the cluster analysis; this, because, while inexpensive, collecting diet information may be troublesome, may require of specific tools (eg, 24-hour diet recalls), and may need context-expert knowledge to ask about foods consumed locally. Similarly, physical activity was not included in the clustering analysis owing to the complexities to measure it. For example, whether physical activity is measured with metabolic equivalents or with questions about days/hours involved in physical activity, not to mention whether it was further disaggregated by moderate or vigorous activity. As argued before for more sophisticated biomarkers, including diet and physical activity profiles could also limit the applicability of the clusters because of the complexities and heterogeneity inherit to these lifestyle cardiometabolic risk factors. Third, we analysed data from NHANES III collected in the late 1980s and early 1990s; that is, data collected roughly 30 years ago. We used NHANES III only because it has liver ultrasound data unlike more recent iterations of the NHANES. Readers should interpret our findings and recommendations considering this limitation. Future work should replicate, verify and advance our work with more contemporary data. Fourth, when missing observations were excluded (because we planned a complete-case analysis), we did not explore the missingness pattern, which is customary in epidemiological research where, if necessary, multiple imputation is conducted and results account for Rubin's rules. The backbone of our work is the unsupervised ML analysis, which needs complete-case data. To the best of our knowledge, there are no techniques to conducted unsupervised ML analyses after multiple imputation. Fifth, with the focus of this work on the identification of unique phenotypes complemented by the mortality analysis, the Cox regressions were not enriched with additional analyses including cause-specific mortality, competing risks, effect modification or interactions. On one hand, some of these analyses would have benefitted from a much larger sample size; on the other hand, some of these analyses depend on both the exposure and outcome variables and in our case, the exposure variable (cluster) was a complex and heterogenous construct of multiple cardiometabolic biomarkers and liver enzymes.

## CONCLUSIONS

There is heterogeneity in patients with NAFLD, whom can be divided into three phenotypes. These phenotypes are either characterised by high levels of anthropometrics, SBP and glucose, or by high levels of cholesterol and liver biomarkers. These phenotypes could guide specific interventions, together with personalised management and follow-up plans thus advancing precision medicine and public health for patients with NAFLD.

**Author affiliations**
[1]Department of Epidemiology and Biostatistics, School of Public Health, Imperial College London, London, UK
[2]Hubert Department of Global Health, Rollins School of Public Health, Emory University, Atlanta, Georgia, USA
[3]CRONICAS Centre of Excellence in Chronic Diseases, Universidad Peruana Cayetano Heredia, Lima, Peru
[4]Universidad Continental, Lima, Peru
[5]School of Medicine Alberto Hurtado, Universidad Peruana Cayetano Heredia, Lima, Peru
[6]Sociedad Científica de Estudiantes de Medicina Cayetano Heredia (SOCEMCH), Universidad Peruana Cayetano Heredia, Lima, Peru
[7]Ontology Engineering Group, Artificial Intelligence Department, Universidad Politécnica de Madrid, Madrid, Spain
[8]Department of Gastroenterology, Hospital Nacional Cayetano Heredia, Lima, Peru
[9]King Faisal Specialist Hospital & Research Centre, Riyadh, Saudi Arabia
[10]Division of Gastroenterology and Hepatology, Department of Medicine, Johns Hopkins University, Baltimore, Maryland, USA
[11]Program in Human Nutrition, Department of International Health, The Johns Hopkins Bloomberg School of Public Health, Baltimore, Maryland, USA

**Acknowledgements** The opinions on this document are those from the authors alone and do not represent the views of the institutions to which they belong.

**Contributors** RMC-L and VG-L conceived the research question. RMC-L pooled and processed the datasets with support from WCG-V. RMC-L conducted the machine learning analysis with support from MC-C. RMC-L conducted the statistical analysis with support from WCG-V. RMC-L drafted the first version of the manuscript with support from WCG-V, MC-C, CA-G, SA and VG-L. WCG-V, MC-C, CA-G, SA and VG-L provided critical input, commented and edited the manuscript. RMC-L, WCG-V, MC-C, CA-G, SA and VG-L approved the submitted version. RMC-L and WCG-V collated, pooled and analysed the data and vouch for the accuracy of the results. RMC-L acts as guarantor.

**Funding** RMC-L is supported by a Wellcome Trust International Training Fellowship (Wellcome Trust 214185/Z/18/Z).

**Competing interests** None declared.

**Patient and public involvement** Patients and/or the public were not involved in the design, or conduct, or reporting, or dissemination plans of this research.

**Patient consent for publication** Not applicable.

**Ethics approval** We analysed individual-level open access data. Human subjects were not directly involved in this work. This work was deemed as of minimal risk. We did not seek approval by an ethics committee because we analysed deidentified data in the public domain available for independent reanalyses.

**Provenance and peer review** Not commissioned; externally peer reviewed.

**Data availability statement** Data are available in a public, open access repository. All NHANES datasets are publicly available and can be downloaded from the official website: https://www.cdc.gov/nchs/nhanes/index.htm All analysis code (Jupyter Notebooks and R Scripts) are available as online supplemental materials.

**ORCID iDs**
Rodrigo M Carrillo-Larco http://orcid.org/0000-0002-2090-1856
Wilmer Cristobal Guzman-Vilca http://orcid.org/0000-0002-2194-8496
Manuel Castillo-Cara http://orcid.org/0000-0002-2990-7090
Vanessa Garcia-Larsen http://orcid.org/0000-0002-0003-1988

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
