## [Reviewer comments · BMJ Open]

ARTICLE DETAILS

TITLE (PROVISIONAL)	Phenotypes of non-alcoholic fatty liver disease (NAFLD) and all-cause mortality: Unsupervised machine learning analysis of NHANES III
AUTHORS	Carrillo-Larco, Rodrigo; Guzman-Vilca, Wilmer Cristobal; Castillo-Cara, Manuel; Alvizuri-Gómez, Claudia; Alqahtani, Saleh; Garcia Larsen, Vanessa

VERSION 1 – REVIEW

REVIEWER	Raja, Ghazala Kaukab Pir Mehr Ali Shah Arid Agr Univ Rawalpindi
REVIEW RETURNED	27-Aug-2022

GENERAL COMMENTS	The study used two designs, cross-sectional and prospective but its not mentioned whether the population was same. For the review of this manuscript my PhD student M. Mobeen Zafar provided assistance.
--

REVIEWER	Ardern, Chris York University, Canada
REVIEW RETURNED	15-Sep-2022

GENERAL COMMENTS	This paper describes an unsupervised machine learning approach to predicting all-cause death in a sample of U.S. adults with ultrasound defined non-alcoholic fatty liver disease (NAFLD). Using data from the U.S. NHANES III (1988-94, n=1 652; 20-74y) with follow-up to Dec. 31st, 2019, a total of three NAFLD (cardiometabolic) phenotypes were identified for further analysis, of which, two demonstrated elevated mortality risk compared to “average”. In general, the paper is well-written and offers somewhat incremental, but important new insight into patient profiles that will serve as a basis for future work. Indeed, identifying phenotypes of NAFLD serves as a basis to examine longitudinal trajectories of disease progression, comorbidities, and subsequent mortality risk in future analyses that may ultimately help to inform advances in precision medicine, such as the timing and need for more aggressive intervention across NAFLD clusters. Despite these strengths, the paper would benefit from additional methodological context, particularly as it relates to the justification for their survival analysis approach. Following are specific areas where additional details and elaboration would help to clarify their analytic approach. Sample:
---

	1. NHANES III: The authors do not explicitly state why they have only made use of NHANES III (1988-1994) data, rather than including more recent (continuous NHANES) cycles to increase their sample? I suspect this is due to the availability and quality of the data and / or time lag between NAFLD and death, but this is not mentioned and should be made explicit. 2. Study Exclusions and Risk of Bias: Although study exclusions and the derivation of the analytic sample are made clear in Supplemental Figure 1 (Participant Flow Chart), it is not clear whether patterns of missingness for hepatic imaging data was explored and confirmed to be NMAR. If not missing at random, this warrants further discussion. If space permits, the article would benefit from an “in-text” description of the supplemental flow chart of study exclusions in the derivation of the analytic sample for the complete case analysis. Analysis: Predictors and Covariates / Confounders: An initial set of 10 possible predictors (established CM risk factors, age, BMI, WC, SBP, plasma glucose, total cholesterol, TG) and liver biomarkers (serum alanine aminotransferase, serum aspartate aminotransferase, and serum gamma gamma glutamyl transferase) were identified and selected with consideration for ease of clinical use / transfer. These variables were exposed to principal components analysis (yielding four PCA explaining over 85% of the variance), following which, a k-means algorithm (unsupervised ML) was then used to identify three unique clusters of NAFLD, and confirmed using other exploratory techniques. In short, the general analysis approach is clear; however, details of specific study predictors and confounders are lacking, and would benefit from elaboration. 3. In particular, it is not clear why physical activity was not considered as a factor in determination of clusters, or as a covariate / confounder in the mortality analysis, given both the authors’ own discussion (P15, lines 11-12) of its therapeutic importance, and the large body of literature supporting the buffering effect of regular PA on mortality risk among those with pre-existing disease in large prospective studies. Likewise, I wonder whether the authors considered further adjustment for i) family history of CVD and ii) alcohol consumption (which despite the exclusion of individuals at the high-end), may still play a role in disease progression and may warrant further investigation. 4. Survival Analysis: It is not clear why the survival analysis was limited to all-cause death. Were models for cause-specific death (e.g. CVD) in a competing risk (sensitivity) analysis considered? At a minimum, the authors should consider reporting the leading causes of death for each phenotype. Given the large age range (20-70 y) and possible sex-difference in mortality risk, it is not clear whether effect modification by age and / or sex is not considered? 5. (P 14, lines 28-32) A minor point, but the rationale for the inclusion of all 10 predictors in a subsequent analysis is not well founded for the reasons the authors articulate. In particular, the inclusion of BMI and WC as covariates should be reconsidered in light of multicollinearity which could contribute to instability of estimates.
--	--

	Results: 6. (P14, Lines 48-9) The authors are encouraged to provide additional context to the metabolic profile of the most prevalent (“average”) phenotype, which represents 83% of the sample. Given that the sample is comprised of those with moderate-to-severe liver disease, reporting absolute risk information in Table 2 in conjunction with their HR estimates would provide much needed additional context. Discussion: 7. It is not surprising that the anthro-SBP-glucose phenotype was most strongly related to all-cause death, as this phenotype includes more risk components, and is consistent with what has been observed by others when examining the (similar) 0 to 5 components of the metabolic syndrome (MetS). While the overlap of the high-risk anthro-SBP-glucose phenotype is acknowledged in the discussion (P 16, line 11), the additive effect of multiple metabolic risk factors is not, and is worth specific mention. Study Limitations: 8. The single time point (baseline 1988-94) assessment precludes the examination of changes in phenotypic patterns; as a result, the authors should exercise caution in their discussion of therapeutic targets and appropriateness of interventions with an acknowledgement of this limitation. This is particularly salient in light of one of the lifestyle recommendations – to lose 7% of weight. While it may be beyond the scope of the current analysis, NHANES includes information on weight history, and could be considered for further sub-analysis. Minor Comments / Typographical Changes: 9. P8, Line 28: “We used time in the study as the time scale in the Cox models”. Could this be restated to better reflect the single baseline measure? 10. P12, line 27: Beyond the prevalence, please add total number of deaths and average follow-up for each phenotype in parentheses. (From table 1, it appears that 41% of the sample died over follow-up). 11. P11, Lines 11-20: In describing the three study clusters (1. Lipid-liver; 2. Anthro-SBP-Glucose, and; 3. Average), the authors may wish to provide an in-text description of the variable ranges and / or thresholds to provide further clinical context. 12. Covariates / confounders (P8, Line 36): It is not clear why smoking status and income were dichotomized, and for income, why poverty-to-income ratio was not considered. Please justify. 13. Descriptive characteristics are presented as boxplots in Figure 1. While informative, for this more clinical audience, the authors may wish to expand Table 1 to include data presented in Supplemental Table 1. Readability may also be improved by transposing columns and rows. 14. P15, Line 38: Although the authors make mention of the therapeutic potential for Vitamin E, the relative importance compared to Vitamins D, B1, B6, or B12, that have also been explored, is not clear.
--	--

	15. Table 1. Consider adding descriptives for all study factors (including additional factors being adjusted for in Model 2 of the multivariable cox regressions). 16. Rather than leaving it for the supplement, consider moving the Kaplan-Meier curve to the main body of the paper. The figure would also benefit from the inclusion of an omnibus chi-square statistic. 17. Ethics statement: Description of analyses being conducted “at the country level” is somewhat misleading. Consider rephrasing to better reflect individual-level analysis of anonymized (open data) that is presented in aggregate. 18. (P8, line 50): Please replace “what” with “that” in sentence “...means what are representative...”
--	--

VERSION 1 – AUTHOR RESPONSE

Reviewer #1

Q1. The study used two designs, cross-sectional and prospective but it's not mentioned whether the population was same.

R1. This has been clarified in the Methods section [p. 04; new text underlined]: *Analysing the same study population, we also conducted a prospective analysis whereby we investigated whether the phenotypes were associated with higher all-cause mortality risk. Overall, for both the cross-sectional and prospective analyses, we used the same study population from the National Health and Nutrition Examination Survey (NHANES) III.*

Reviewer #2

Q1. NHANES III: The authors do not explicitly state why they have only made use of NHANES III (1988-1994) data, rather than including more recent (continuous NHANES) cycles to increase their sample? I suspect this is due to the availability and quality of the data and / or time lag between NAFLD and death, but this is not mentioned and should be made explicit.

R1. We clarified this with a new sentence in the Method section [p. 05]: *We used NHANESS III, and no more recent iterations of this survey, because NHANESS III is the only with liver ultrasound data which provide high-quality information for NALFD research.*

This matter was also discussed as a limitation in the Discussion section [p. 14]: *Third, we analysed data from NHANES III collected in the late 1980s and early 1990s; that is, data collected roughly 30 years ago. We used NHANES III only because it has liver ultrasound data unlike more recent iterations of the NHANES. Future work should replicate, verify, and advance our work with more contemporary data.*

Q2. Study Exclusions and Risk of Bias: Although study exclusions and the derivation of the analytic sample are made clear in Supplemental Figure 1 (Participant Flow Chart), it is not clear whether patterns of missingness for hepatic imaging data was explored and confirmed to be NMAR. If not missing at random, this warrants further discussion. If space permits, the

article would benefit from an “in-text” description of the supplemental flow chart of study exclusions in the derivation of the analytic sample for the complete case analysis.

R2. No, missing patterns for hepatic imaging data were not explored nor confirmed to be missing at random. This was acknowledged in the Methods section [p. 05]: *In so doing, we excluded missing observations in liver ultrasound data; missing patterns in this variable were not explored.*

Acknowledging the reviewer’s concern, we elaborated about this in the limitations [p. 16]: *Fourth, when missing observations were excluded (because we planned a complete-case analysis) we did not explore the missingness pattern which is customary in epidemiological research where, if necessary, multiple imputation is conducted and results account for Rubin’s rules. The backbone of our work is the unsupervised machine learning analysis which needs complete-case data. To the best of our knowledge, there are no techniques to conducted unsupervised machine learning analyses after multiple imputation.*

If the reviewer and editors kindly allow, we would rather not expand the description about the derivation of the analytic ample in the manuscript. We would like to keep the manuscript succinct and focused on the main methods and results (i.e., cluster analysis and long-term outcomes). A detailed description about the derivation of the analytic sample is available in the Supplementary Materials, both as text and as a flowchart.

Q3. In particular, it is not clear why physical activity was not considered as a factor in determination of clusters, or as a covariate / confounder in the mortality analysis, given both the authors’ own discussion (P15, lines 11-12) of its therapeutic importance, and the large body of literature supporting the buffering effect of regular PA on mortality risk among those with pre-existing disease in large prospective studies. Likewise, I wonder whether the authors considered further adjustment for i) family history of CVD and ii) alcohol consumption (which despite the exclusion of individuals at the high-end), may still play a role in disease progression and may warrant further investigation.

R3. As it was discussed for diet, which was not included in the analysis either, we included in the limitations some lines to discuss the lack of physical activity data in the analysis [p. 16; new text underlined]: *Similarly, physical activity was not included in the clustering analysis owing to the complexities to measure it. For example, whether physical activity is measured with metabolic equivalents (METs) or with questions about days/hours involved in physical activity, not to mention whether it was further disaggregated by moderate or vigorous activity. As argued before for more sophisticated biomarkers, including diet and physical activity profiles could also limit the applicability of the clusters because of the complexities and heterogeneity inherit to these lifestyle cardiometabolic risk factors.*

From a classic epidemiological definition, regression models can be adjusted for confounders which are variables associated with both the exposure and outcome. In our regression analysis, the exposure variable is a ‘complex’ construct derived from the unsupervised machine learning analysis accounting for multiple laboratory biomarkers (not just cardiometabolic risk factors but also liver enzymes). Whether family history of cardiovascular diseases would be associated with the cluster construct is debatable.

We did not include alcohol in the analysis following the same logic as to why we did not include diet profiles. This was specified in the limitations [p. 16; new text underlined]: *In this line, although diet is a key variable in NAFLD, we did not include diet-based variables (including alcoholic beverages) in the cluster analysis; this, because, while inexpensive, collecting diet information may be troublesome, may require of specific tools (e.g., 24-hour diet recalls), and may need context-expert knowledge to ask about foods consumed locally.*

Q4. Survival Analysis: It is not clear why the survival analysis was limited to all-cause death. Were models for cause-specific death (e.g. CVD) in a competing risk (sensitivity) analysis considered? At a minimum, the authors should consider reporting the leading causes of death for each phenotype. Given the large age range (20-70 y) and possible sex-difference in mortality risk, it is not clear whether effect modification by age and / or sex is not considered?

R4. The reason for only working with all-cause mortality was explained on page 06: *Because a priori we did not know the number of clusters or their underlying profiles, we focused on all-cause mortality rather than on a specific cause of death.* These lines were complemented with new text also on page 06: *For example, had the unsupervised analysis suggested many clusters, it would have compromised the confidence intervals from the regressions if the exposure variable was divided in several clusters or groups.*

The focus of the work was the clustering analysis to identify different phenotypes, with the mortality analysis as a complement. Additional mortality analyses, such as specific-cause mortality or competing risk analyses, were not considered. Please, also bear in mind that the NHANES mortality data does not include a comprehensive description of the cause of death. In fact, the underline cause of death is only available in nine groups (e.g., diseases or heart, malignant neoplasms, chronic lower respiratory diseases).

Effect modification or interactions were not considered. First, these concepts depend on both the exposure and the outcome and as discussed before, the exposure is a 'complex' construct of heterogenous biomarkers including cardiometabolic and liver enzymes. Second, the sample size (~1,600 people) was rather small to consider additional biostatistical analyses. These arguments were discussed amongst the limitations [p. 16]: *Fifth, with the focus of this work on the identification of unique phenotypes complemented by the mortality analysis, the Cox regressions were not enriched with additional analyses including cause-specific mortality, competing risks, effect modification or interactions. On one hand, some of these analyses would have benefitted from a much larger sample size; on the other hand, some of these analyses depend on both the exposure and outcome variables and in our case, the exposure variable (cluster) was a complex and heterogenous construct of multiple cardiometabolic biomarkers and liver enzymes.*

Q5. (P 14, lines 28-32) A minor point, but the rationale for the inclusion of all 10 predictors in a subsequent analysis is not well founded for the reasons the authors articulate. In particular, the inclusion of BMI and WC as covariates should be reconsidered in light of multicollinearity which could contribute to instability of estimates.

R5. Lines 28-32 on 14 of the original submission describe the results of the regression for model 2. As described both in the text (Methods) and in the footnote to Table 2, regression model 2 included sex, smoking status, years of education, and household income. That is, model 2 (and neither does model 1) did not include body mass index or waist circumference.

We are clarifying this potential confusion with these lines in the footnote to Table 2: *The exposure variable for all models (crude, model 1, and model 2) was a three-level variable derived from the cluster analysis; in other words, the cluster analysis derived a new variable with three levels which we called phenotypes (Average, Lipid-Liver, and Anthro-SBP-Glucose) whose profiles were described in Figure 1. Neither of the three models included the predictors used to derive the clusters or phenotypes. The predictors were used to derive and characterize the phenotypes only (not to adjust the models).*

Q6. (P14, Lines 48-9) The authors are encouraged to provide additional context to the metabolic profile of the most prevalent (“average”) phenotype, which represents 83% of the sample. Given that the sample is comprised of those with moderate-to-severe liver disease, reporting absolute risk information in Table 2 in conjunction with their HR estimates would provide much needed additional context.

R6. The metabolic profile of the Average phenotype is described in Figure 1 and Supplementary Table 1 and Supplementary Table 2. The three phenotypes are described consistently with the same level of detail in the manuscript. Although the Average phenotype is the most frequent phenotype, it is not the core of our narrative which focused on the other two phenotypes with clearer profiles.

Table 2 was complemented to also show the absolute number of deaths by group, the total follow-up duration by group, as well as the incidence rate per 1,000 person-years at risk by group including the associated 95% confidence interval. Please, refer to Table 2 in the updated manuscript for further details. These new results were also acknowledged in the Method section [p. 08].

Q7. It is not surprising that the anthro-SBP-glucose phenotype was most strongly related to all-cause death, as this phenotype includes more risk components, and is consistent with what has been observed by others when examining the (similar) 0 to 5 components of the metabolic syndrome (MetS). While the overlap of the high-risk anthro-SBP-glucose phenotype is acknowledged in the discussion (P 16, line 11), the additive effect of multiple metabolic risk factors is not and is worth specific mention.

R7. This was discussed on page 15: *It is worth noting that, even though on average the Anthro-SBP-Glucose phenotype had high levels of anthropometrics, SBP and glucose, it is not the same as a dichotomous variable whereby people with high anthropometrics, SBP and glucose were on one or the other category of the dichotomous variable (similar to the definition of metabolic syndrome); nevertheless, the additive effect of the anthropometrics, SBP and glucose should be noted and could potential explain the higher mortality.*

Q8. The single time point (baseline 1988-94) assessment precludes the examination of changes in phenotypic patterns; as a result, the authors should exercise caution in their discussion of therapeutic targets and appropriateness of interventions with an acknowledgement of this limitation. This is particularly salient in light of one of the lifestyle recommendations – to lose 7% of weight. While it may be beyond the scope of the current analysis, NHANES includes information on weight history, and could be considered for further sub-analysis.

R8. This limitation was discussed on page 16: *Third, we analysed data from NHANES III collected in the late 1980s and early 1990s; that is, data collected roughly 30 years ago. We used NHANES III only because it has liver ultrasound data unlike more recent iterations of the NHANES. Readers should interpret our findings and recommendations considering this limitation. Future work should replicate, verify, and advance our work with more contemporary data.*

Q9. P8, Line 28: “We used time in the study as the time scale in the Cox models”. Could this be restated to better reflect the single baseline measure?

R9. This sentence was modified to [p. 07; new text underlined]: *We used time in the study (difference between when the one-off baseline evaluation occurred and the endpoint) as the time scale in the Cox models.*

Q10. P12, line 27: Beyond the prevalence, please add total number of deaths and average follow-up for each phenotype in parentheses. (From table 1, it appears that 41% of the sample died over follow-up).

R10. Please, refer to Table 2 in the updated manuscript where we have included this information. We also elaborated about this in our sixth comment above.

Q11. P11, Lines 11-20: In describing the three study clusters (1. Lipid-liver; 2. Anthro-SBP-Glucose, and; 3. Average), the authors may wish to provide an in-text description of the variable ranges and / or thresholds to provide further clinical context.

R11. If the reviewer and editors allow, we would kindly request not to provide in-text description of the variable ranges. All information is available as figures and Supplementary Tables; the interested reader can find these details seamlessly. There are no thresholds in any of the variables used to define or describe the phenotypes; that is, all variables were treated as numeric.

Q12. Covariates / confounders (P8, Line 36): It is not clear why smoking status and income were dichotomized, and for income, why poverty-to-income ratio was not considered. Please justify.

R12. Both variables, smoking and income, were available in such form in the original dataset.

Q13. Descriptive characteristics are presented as boxplots in Figure 1. While informative, for this more clinical audience, the authors may wish to expand Table 1 to include data presented in Supplemental Table 1. Readability may also be improved by transposing columns and rows.

R13. Table 1 was complemented with the information originally presented in Supplementary Table 1 only. Supplementary Table 1 was therefore removed from the Supplementary Materials because it did not provide new results. The numeration of the Supplementary Materials was updated accordingly.

Q14. P15, Line 38: Although the authors make mention of the therapeutic potential for Vitamin E, the relative importance compared to Vitamins D, B1, B6, or B12, that have also been explored, is not clear.

R14. The choice of Vitamin E was an example, and the fact that we did not highlight other vitamins was elaborated [p. 14; new text underlined]: *For example, vitamin E has been proposed as a potential therapy for NAFLD, because it may improve non-alcoholic steatohepatitis. However, systematic reviews and meta-analysis of randomised trials have shown inconsistent evidence to recommend vitamin E for NAFLD; moreover, potential adverse effects of vitamin E should be taken into consideration. Although other vitamins may play a role as well, to the best of our knowledge, their effect has not been summarized in meta-analysis of randomized control trials (high level of evidence) and is less elaborated in clinical guidelines.*

Q15. Table 1. Consider adding descriptives for all study factors (including additional factors being adjusted for in Model 2 of the multivariable cox regressions).

R15. If the reviewers and editors kindly allow, we would rather not include such variables in Table 1. Table 1 summarizes the variables included in the development of the three clusters. Describing other variables in terms of the three phenotypes is beyond the scope of this work: define and characterize clusters, and complementary estimation of long-term outcomes (mortality risk). Describing, for example, education level of income across the three phenotypes would imply we are also studying health inequalities or socio-demographic determinants of health, which is not the case.

Q16. Rather than leaving it for the supplement, consider moving the Kaplan-Meier curve to the main body of the paper. The figure would also benefit from the inclusion of an omnibus chi-square statistic.

R16. Moving the Kaplan-Meier curve to the main manuscript will draw attention to the mortality analysis, which is not the main narrative of our work. Kaplan-Meier curve and other results are available in the Supplementary Materials.

Q17. Ethics statement: Description of analyses being conducted “at the country level” is somewhat misleading. Consider rephrasing to better reflect individual-level analysis of anonymized (open data) that is presented in aggregate.

R17. The ethics statement was changed as suggested [p. 09]: *We analysed individual-level open access data.*

Q18. (P8, line 50): Please replace “what” with “that” in sentence “...means what are representative...”

R18. Changed as suggested.

VERSION 2 – REVIEW

REVIEWER	Ardern, Chris York University, Canada
REVIEW RETURNED	07-Nov-2022

GENERAL COMMENTS	Please note: Akinkunle Oye-Somefun (York University) and I completed this review jointly. We thank the authors for their thoughtful responses and revisions. The authors have responded to all major comments and adequately addressed all others. This includes providing additional context to the study rationale - to demonstrate the utility of unsupervised ML ("precision medicine") with NHANES data in patients with NAFLD. We agree with the reviewer response that further details on the derivation of the analytic sample are not necessary in the manuscript body, and have no further comments. We recommend accepting the manuscript in its current form.
---